# The use of virtual reality in screening for preclinical Alzheimer's disease: A scoping review protocol

**Yuan Tian[1]\*, Maneesh V. Kuruvilla[2‡], Mira Park[1‡]**

**1** School of Information and Communication Technology, University of Tasmania, Hobart, Tasmania, Australia, **2** Wicking Dementia Research and Education Centre, University of Tasmania, Hobart, Tasmania, Australia

‡ MVK and MP are joint senior authors on this work.
\* y.tian@utas.edu.au

**Data Availability Statement:** All relevant data will be made publicly available when the study is completed and published.

## Abstract

### Introduction

Preclinical Alzheimer's disease (AD) represents the earliest phase of AD, often years before the onset of mild cognitive impairment (MCI). There is a pressing focus on identifying individuals in the preclinical AD phase to alter the trajectory or impact of the disease potentially. Increasingly, Virtual Reality (VR) technology is being used to support a diagnosis of AD. While VR technology has been applied to the assessment of MCI and AD, studies about how best to utilize VR as a screening tool for preclinical AD are limited and discordant. The objectives of this review are to synthesize the evidence pertaining to the use of VR as a screening tool for preclinical AD as well as to identify factors that need to be considered when utilizing VR to screen for preclinical AD.

### Methods and analysis

The methodological framework proposed by Arksey and O'Malley (2005) will be introduced to guide the conduction of the scoping review, and Preferred Reporting Items for Systematic Reviews and Meta-Analyses extension for scoping reviews (PRISMA-ScR) (2018) will be used to organize and structure the review. PubMed, Web of Science, Scopus, ScienceDirect and Google Scholar will be used to search for literature. Obtained studies will be screened for eligibility based on predefined exclusion criteria. A narrative synthesis of eligible studies will be performed, after tabulating the extracted data from existing literature, to answer the research questions.

### Ethics and dissemination

Ethical approval is not required for this scoping review. Findings will be disseminated through conference presentations, publication in a peer-reviewed journal, and discussions among professional networks in the research domain combining neuroscience and information and communications technology (ICT).

**Funding:** The authors received no specific funding for this work.

**Competing interests:** The authors have declared that no competing interests exist.

## Registration details

This protocol has been registered on Open Science Framework (OSF). Relevant materials and potential following updates are available at https://osf.io/aqmyu.

## Introduction

There is an increasing emphasis on focusing research efforts on screening for MCI and AD as early as possible [1, 2]. This is largely down to the fact that AD has a long preclinical phase—often years to decades—when cognitive impairments are largely undetectable, providing a potential window of opportunity to intervene [3, 4]. Increasingly, VR technologies are being used to identify cognitive markers that might help screen for preclinical AD in light of emerging evidence from the field of navigation neuroscience [5, 6]. Spatial navigation is supported by structures in the medial temporal lobe, most notably the hippocampus and entorhinal cortex [7–9]. The entorhinal cortex, in particular, shows some of the earliest signs of AD pathology [10], making tests of navigation an attractive option to assess preclinical AD over most established diagnostic tests of episodic memory [11]. Tests of navigation often require larger-scale environments, which can be fully simulated in VR, necessitating minimal testing space [12, 13]. The environments need to appear realistic but also allow for a degree of control to test specific aspects of navigation supported by brain areas in question [14, 15]. Immersive VR, in particular, provides an opportunity to add to the realism of the environment thereby providing a more accurate representation of the space being tested and allowing for more ecologically valid studies to be conducted. The disparate requirements needed to integrate digital and physical space are best supported by VR technologies, which is why they are being increasingly utilized to validate navigation screening tests. There is, therefore, a need to synthesize information about how VR technologies can be harnessed to develop preclinical AD cognitive markers. This has the potential to help inform research across multiple fields including neuroscience, psychology and ICT [16].

This review aims to answer two research questions:

1. What is known about the use and viability of VR as a preclinical AD screening tool?

2. What factors need to be considered when utilizing VR tools to screen for preclinical AD?

## Materials and methods

This protocol details specific methodological steps to conduct the scoping review. The proposal of this protocol is based on the Joanna Briggs Institute's (JBI) Reviewer's Manual [17] and PRISMA-ScR [19]. JBI Reviewer's Manual presents the methodology framework for conducting a scoping review proposed by Arksey and O'Malley [18]. The framework comprises 5 compulsory stages: 1) identifying the research question(s), 2) identifying relevant studies, 3) selecting studies, 4) charting the data, 5) collating, summarizing, and reporting the results. The five stages will be accomplished throughout the process of finalizing the review article. In addition, the checklist of PRISMA-ScR [19] lists all the items required by different sections of a scoping review, and it will be utilized to organize and structure the review.

### Stage 1: Identifying the research question

VR has generated promising results with high sensitivity and specificity in the diagnosis of MCI and dementia due to AD [20, 21], but preliminary searches reveal limited adoption of VR as a screening tool for preclinical AD. Additionally, there is no existing review synthesizing the

literature about using VR as a preclinical AD screening, thus, it is uncertain how VR technology is applied in preclinical AD detection and whether this technology is a viable tool. Furthermore, factors which may affect the success of a VR-related preclinical AD screening tool are also unclear. Clarifying these factors will have significant implications for interdisciplinary domains such as neuroscience and computer science in developing future VR-based AD screening tools. Based on preliminary searches and discussions within the research team, two research questions are identified:

1. What is known about the use and viability of VR as a preclinical AD screening tool?

2. What factors need to be considered when utilizing VR tools to screen for preclinical AD?

## Stage 2: Identifying relevant studies

In order to identify relevant studies that address the chosen research questions, the following definitions of VR and preclinical AD have been selected.

VR, as defined by Mazuryk and Gervautz [22], refers to a virtual environment that creates sensory impressions that can be experienced through human senses. The authors further parcel out VR into three broad categories based on the level of immersion on offer, including Desktop VR, Fish Tank VR and Immersive VR. The use of the level of immersion, i.e., a sense of presence in a virtual environment, as a classifier of the VR taxonomy is also supported by other studies, but the terminology of the specific types of immersive VR vary [23–25]. For consistency, we will refer to those VR types as non-immersive VR, semi-immersive VR and fully immersive VR respectively. Non-immersive VR is the most accessible form of VR where content is displayed on a monitor and the viewer receives limited sensory input. The combination of a monitor, a mouse and a keyboard is usually adopted by a non-immersive-VR system [26]. This is akin to playing a computer game. As for semi-immersive VR, Mandal [27] illustrated that head tracking was usually supported by this VR system, and with the motion parallax effect, users could have a stronger feeling of presence than in non-immersive VR. Examples of semi-immersive VR are a flight simulator and watching a movie in the cinema while wearing 3D glasses [22]. Fully immersive VR represents the gold standard of VR systems and is able to generate environments that actively update along with the user's physical location and heading direction. Additionally, these systems may offer a variety of other sensory interfaces such as audio and haptic information. Immersive environments are characterised by users wearing a head-mounted display such as the Oculus Rift, Meta Quest or PlayStation VR as they physically explore virtual spaces.

The definition of preclinical AD will be kept relatively broad for a couple of reasons. One, there is a lack of consensus on what factors translate from asymptomatic, preclinical risk to symptomatic, clinical impairments thus making it challenging to produce a specific definition of preclinical AD. Two, the proposed change in definition to a focus on measures of underlying pathophysiology is a relatively recent one proposed in 2011 based on recommendations from the National Institute on Aging-Alzheimer's Association workgroups on diagnostic guidelines for Alzheimer's disease [28]. Applying a definition based purely on pathophysiology (e.g., amyloid/tau in the brain and/or cerebrospinal fluid (CSF)) may exclude earlier relevant papers from this scoping review. In general, there are two ways of classifying individuals that fall into the preclinical AD category. According to criteria set out by the 2014 International Working Group-2 (IWG-2), individuals with preclinical AD are defined as being asymptomatic with the presence of amyloidopathy or tauopathy in the brain or in the CSF [29]. Alternately, preclinical AD can be separated along the lines of high v. low risk based on factors such as genetics, brain atrophy, modifiable risk factors and even subjective cognitive decline [30].

**Table 1. PCC framework for identifying key concepts of the scoping review.**

| PCC component | Definition in this study |
|---|---|
| Population | Eligible studies should include participants with preclinical AD. Within the scope of this review, those individuals should be clinically asymptomatic with the presence of amyloidopathy or tauopathy in the brain and/or CSF, or have high-risk factors previously outlined. |
| Concept | Eligible studies should be screening-related. Screening is defined as "the testing or examining of a large number of people or things for disease, faults, etc." [31]. In this case, eligible studies should include information about participants who have been screened for preclinical AD. |
| Context | Eligible studies should apply VR technology in their study methods. VR is a technology that has the capability to "make that (virtual) world in the window look real, sound real, feel real, and respond realistically to the viewer's actions" [22]. In this scoping review, no specific requirement on the type of VR is proposed, thus, all non-immersive VR, semi-immersive VR and immersive VR will be included. |

For the purposes of this scoping review, we will select all papers that make use of a preclinical AD cohort and then clearly identify the basis on which preclinical AD has been defined in our reporting.

Relevant studies will be identified by adopting a customized search strategy for the two research questions. The search strategy described below has been developed in consultation with a librarian at the University of Tasmania. The Population-Concept-Context (PCC) Framework recommended by the JBI [17] has been used to identify key concepts and search terms when establishing a search strategy. The PCC framework for this scoping review study is shown in Table 1, and a template employed to select relevant studies in PubMed is presented in Table 2. Based on the template, Table 3 exemplifies a specific search strategy used in PubMed. Another two databases, including Web of Science and Scopus, will also be utilized for searching the literature. Meanwhile, Google Scholar and ScienceDirect will be used to identify additional relevant studies. Due to the different searching features of the aforementioned databases, the template presented in Table 2 is expected to adjust before applying to the corresponding databases, however, the key concepts will be consistent.

## Stage 3: Selecting studies

Study selection will involve 4 steps. First, the obtained search results from Stage 2 will be imported to Endnote (a reference manager software). Second, with the assistance of Endnote, duplicates will be removed. The third step is to read titles and abstracts in order to exclude

**Table 2. Template used to identify key concepts and search terms.** Key concepts are proposed by referring to the research questions. Free text terms are based on synonyms and terminology, and controlled vocabulary terms refer to MeSH terms in PubMed. "*" stands for a wildcard, which can be replaced by any character(s).

| | Concept 1 | Concept 2 | Concept 3 |
|---|---|---|---|
| Key concepts | Virtual reality | Preclinical Alzheimer's disease | Screening |
| Free text terms / natural language terms | Virtual | Alzheimer* | Screening |
| | Game | Preclinical | Detect* |
| | Games | Subjective cognitive decline | Diagnos* |
| | Computer | | Assess* |
| | | | Evaluat* |
| | | | Testing |
| | | | Analysing |
| | | | Genetic risk |
| Controlled vocabulary terms / subject terms | Virtual reality | Alzheimer disease | Mental Status and Dementia Tests |

**Table 3. Search strategy example in PubMed as of 09 January 2023.**

| Search number | Query | Results |
|---|---|---|
| 1 | ((((Virtual reality[MeSH Terms]) OR (Virtual[Title/Abstract])) OR (Game[Title/Abstract])) OR (Games[Title/Abstract])) OR (Computer[Title/Abstract]) | 371,262 |
| 2 | (((Alzheimer disease[MeSH Terms]) OR (Alzheimer*[Title/Abstract])) OR (Preclinical[Title/Abstract])) OR (Subjective cognitive decline[Title/Abstract]) | 332,337 |
| 3 | (((((((Mental Status and Dementia Tests[MeSH Terms]) OR (Screening[Title/Abstract])) OR (Detect*[Title/Abstract])) OR (Diagnos*[Title/Abstract])) OR (Assess*[Title/Abstract])) OR (Evaluat*[Title/Abstract])) OR (Testing[Title/Abstract])) OR (Analysing[Title/Abstract])) OR (Genetic risk[Title/Abstract]) | 10,939,023 |
| 4 | #1 AND #2 AND #3 | 2,568 |

studies that meet exclusion criteria (as shown in Table 4). Finally, the full text of the studies shortlisted in step 3 will be read and examined to exclude studies that meet exclusion criteria.

The process of identifying eligible studies will be conducted simultaneously by multiple reviewers given the large volume of potentially relevant records. To facilitate the accuracy of this process, a small sample of records will initially be selected to pilot the process of using inclusion and exclusion criteria to shortlist relevant records. Any disagreements that arise as part of the pilot will be resolved through discussion. Once consensus has been reached among reviewers, all remaining records will be divided across reviewers and assessed for eligibility. Records that are selected for eligibility will be verified by a senior member of the research team. The results of this stage will be presented in a flow chart (shown in Fig 1), aligned with the PRISMA-ScR statement [19].

## Stage 4: Charting the data

The charting process will initially be piloted by reviewers with domain expertise in ICT and neuroscience fields respectively as well as a third reviewer with domain general expertise. The pilot will be conducted on three papers. Any disagreements that arise as part of the pilot will be resolved through discussion. Once consensus has been reached among reviewers, one reviewer will chart data from all remaining papers followed by verification by a second reviewer. Following the JBI Reviewer's Manual [17], information items (as shown in Table 5) will be extracted initially. When certain items are difficult to identify, the authors of eligible articles will be contacted to obtain the information.

## Stage 5: Collating, summarizing and reporting the results

To address the two review questions, a narrative synthesis approach will be utilised. This will facilitate the analysis of each included evidence source, evaluating them from both neuroscience and ICT perspectives, given the interdisciplinary nature of the research questions. In addition, summary tables will be generated to present the categorized data, the content of

**Table 4. Exclusion criteria.**

| Item No. | Exclusion criteria |
|---|---|
| 1 | Records that do not address all three concepts identified in the PCC framework |
| 2 | Records without abstracts or where full-text versions cannot be obtained. |
| 3 | Secondary and/or non-peer-reviewed literature such as conference posters, reviews, opinion pieces, dissertations and reviews. |

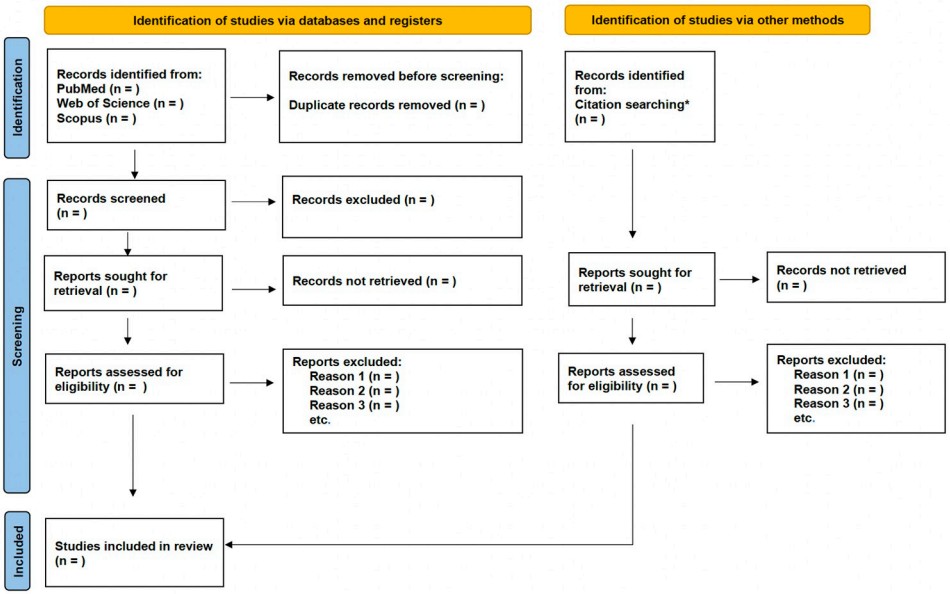

**Fig 1. Flow diagram for the scoping review adapted from the PRISMA 2020 flow diagram [32].**

which will be aligned to answer the research questions. Appropriate figures will also be used to present the findings. Limitations of current existing studies and potential research directions will be identified to inform future research. Moreover, the PRISMA-ScR checklist [19] will guide the progress of result collating, summarizing and reporting.

## Ethics and dissemination

No human participants will be involved in this study, and no data will be collected directly from human participants. The scoping review will only be based on published data, thus,

**Table 5. Definitions of information items to be extracted initially [17].**

| No. | Item | Definition |
|---|---|---|
| 1 | Author(s) | Author(s) of the article |
| 2 | Year | Year of the article published |
| 3 | Origin | Place where the study was conducted or the article published |
| 4 | Aims | Objectives of the study |
| 5 | Study population and sample size | Group characteristics (e.g. individuals with preclinical AD v. those diagnosed with AD) and related sample sizes. |
| 6 | Methodology | The methodology adopted by the study to achieve its aims or to explore the answers to its research questions |
| 7 | Type of VR used | Non-immersive, semi-immersive, fully-immersive |
| 8 | Evidence of preclinical AD | Amyloidopathy or tauopathy detected PET imaging/CSF measures, genetic risk factors etc. |
| 9 | Outcome measures / results | The main result(s) of the screening task(s) conducted by the study, e.g., sensitivity and specificity, if applicable. |
| 10 | Key findings that relate to the scoping review questions | Finding(s) of the study which is/are related to two proposed scoping review questions, for example, successful screening task(s) and how it was designed or the possible factors/reasons behind unsuccessful task(s) |

ethical approval is not required. Findings of this study will be disseminated through conference presentations, publication in a peer-reviewed journal and discussions among professional networks in the research domain combining neuroscience and Information Communications and Technology.

## Strengths and limitations of this study

This scoping review is the first to collate evidence on how VR tools are used to screen for preclinical AD and to explore the corresponding factors that may affect the viability of the VR tools for preclinical AD screening. This scoping review utilises a broad definition of VR, including non-immersive VR (e.g., a screen-based VR), semi-immersive VR (e.g., a flight simulator), and immersive VR (e.g., head-mounted display), which facilitates the recognition of current research directions across the three types of VR methods and the identification of underlying research gaps. This scoping review will be conducted by a multi-disciplinary research team with researchers from both ICT and neuroscience. Aligned with the PCC framework, this protocol lists explicit inclusion and exclusion criteria for the studies to be reviewed. Following the guidelines adopted to instruct this review, the quality of the evidence collected will not be critically appraised, thus, the strength and the weakness of the evidence collected by this study about the viability of adopting the VR technology as a preclinical screening tool will not be discussed.

## Conclusion

This scoping review will collate evidence pertaining to how VR tools are used to screen for preclinical AD and to explore the corresponding factors that may affect the viability of the VR tools for preclinical AD screening. It aims to facilitate the recognition of current research directions about VR-based preclinical AD screening tools and identify potential gaps for future studies.

## Supporting information

**S1 Checklist. Preferred reporting items for systematic reviews and meta-analyses extension for scoping reviews (PRISMA-ScR) checklist adapted for this protocol.**
(DOCX)

## Acknowledgments

We would like to thank Michaela Venn, the learning and research librarian at the University of Tasmania, for her input in developing the search strategy. Additionally, we would like to thank Dr. Winyu Chinthammit from the University of Tasmania for his suggestions on this work.

## Author Contributions

**Conceptualization:** Yuan Tian, Maneesh V. Kuruvilla.

**Investigation:** Yuan Tian.

**Methodology:** Yuan Tian, Maneesh V. Kuruvilla.

**Supervision:** Maneesh V. Kuruvilla, Mira Park.

**Writing – original draft:** Yuan Tian.

**Writing – review & editing:** Maneesh V. Kuruvilla, Mira Park.

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
