## [Editor Report · Decision Letter 0]

31 Aug 2022

PONE-D-22-23396

The use of virtual reality in screening for preclinical Alzheimer’s disease: a scoping review protocol

PLOS ONE

Dear Dr. Tian,

Thank you for submitting your manuscript to PLOS ONE. After careful consideration, we have decided that your manuscript does not meet our criteria for publication and must therefore be rejected.

Specifically: The authors refer to this work as a review, but to me it is not. Indeed it is more a bibliographic study about publications in a specific topic.

The paper is unsound for PlosOne.

I am sorry that we cannot be more positive on this occasion, but hope that you appreciate the reasons for this decision.

Kind regards,

António M. Lopes, PhD

Academic Editor

PLOS ONE

- - - - -

---

## [Author Response · Author response to Decision Letter 0]

27 Sep 2022

Dear PLOS One,

For your information, this manuscript has been approved to be reconsidered after an appeal. Thanks so much. Here is the corresponding response to the specific comments. 

1. We, the authors, refer to the manuscript as a scoping review protocol (as stated as early on as in the title) and not a scoping review. We believe that this may be the main source of misunderstanding between us and the editor.

2. Having read the PLOS One 'Criteria of Publication,' we understand that reviews, on their own, will not be considered for publication. Instead protocols, including scoping review protocols, will be considered for publication by PLOS One. Indeed, one of the co-authors, Dr Kuruvilla, has a scoping review protocol that is currently underdoing peer review at PLOS One (PONE-D-22-08684). PLOS One promotes the publication of scoping review protocols to ensure academic transparency, which is why we have selected this Journal for our work. 

Thank you.

Best regards,

Yuan Tian

---

## [Decision Letter · Decision Letter 1]

23 Dec 2022

PONE-D-22-23396R1

The use of virtual reality in screening for preclinical Alzheimer’s disease: a scoping review protocol

PLOS ONE

Dear Dr. Tian,

Thank you for submitting your manuscript to PLOS ONE. After careful consideration, we feel that it has merit but does not fully meet PLOS ONE’s publication criteria as it currently stands. Therefore, we invite you to submit a revised version of the manuscript that addresses the points raised during the review process.

Please see the comments from two reviewers below. Reviewer 1 has raised a couple of important questions about the definitions used, and thus the intended scope of the study. Reviewer 2 has provided several detailed comments on the manuscript as a whole, some of which overlap with Reviewer 1's concerns. We invite you to consider these comments carefully.

We look forward to receiving your revised manuscript.

Kind regards,

Hanna Landenmark

Staff Editor

PLOS ONE

Journal Requirements:

Additional Editor Comments (if provided):

Reviewers' comments:

Reviewer's Responses to Questions

**Comments to the Author**

1. Does the manuscript provide a valid rationale for the proposed study, with clearly identified and justified research questions?

Reviewer #1: Partly

Reviewer #2: Yes

2. Is the protocol technically sound and planned in a manner that will lead to a meaningful outcome and allow testing the stated hypotheses?

Reviewer #1: Partly

Reviewer #2: Yes

3. Is the methodology feasible and described in sufficient detail to allow the work to be replicable?

Reviewer #1: Yes

Reviewer #2: Yes

4. Have the authors described where all data underlying the findings will be made available when the study is complete?

Reviewer #1: Yes

Reviewer #2: Yes

5. Is the manuscript presented in an intelligible fashion and written in standard English?

Reviewer #1: Yes

Reviewer #2: Yes

6. Review Comments to the Author

You may also provide optional suggestions and comments to authors that they might find helpful in planning their study.

Reviewer #1: Thank you for your submission.

I would like to see a more rigorous definition of VR in the introduction. For example, clarification on what you mean by 'screen-based' VR - there are articles that use 'VR' to describe participants looking at a monitor where physical movement (e.g. head rotation) does not correspond to digital movement updating. If VR is looking at a screen without physical-digital correspondence, then almost all cognitive papers examining preclinical AD can be included. This then becomes a scoping review protocol of preclinical AD, not VR.

In a similar vein, I would like to see a more rigorous definition of preclinical AD. Do these individuals have symptoms - should we consider them as having evidence of pathology or just modifiable/non-modifiable risk factors? This could further extend to detail about what biomarkers are considered sufficient evidence of preclinical AD. For example, Ab and tau positive PET/CSF evidence is stronger and more direct evidence of AD pathology than the presence of 1 or more ApoE4 alleles.

Reviewer #2: The authors present a protocol that will be used to produce a scoping review. The review is focused on the use of Virtual Reality (VR) technology as a screening tool for preclinical Alzheimer's disease (AD). The objectives of the review are to summarize the evidence for using VR for preclinical AD screening, and identify factors to consider when using VR for this purpose. The review will use the Arksey and O’Malley methodological framework and the PRISMA-ScR extension for scoping reviews. It will search for literature using PubMed, Web of Science, Scopus, ScienceDirect, and Google Scholar. Eligible studies will be screened by three reviewers using predefined exclusion criteria.

I support the authors well-defined protocol, as it addresses an important gap in the existing literature. I encourage the publication of the protocol, as it aligns with the growing movement towards transparency in research. However, I have some comments to the protocol

Introduction

“Spatial navigation is supported by brain areas that show some of the earliest signs of AD pathology”.

I would add a sentence to spoke out these brain areas, at least mentioning the medial temporal lobe for example.

“Tests of navigation often require larger-scale environments than can be set up and assessed in a small testing space [13] [14].”

I would change this sentence slightly to point out that there is virtual reality that simulate the environments as the small testing space here seems to refer to a simple tablet or a monitor. Something like “Tests of navigation often larger-scale environments, which can be fully simulated in VR, allowing for minimal testing space”

“The environments need to appear realistic but also allow for a degree of control to test specific aspects of navigation supported by brain areas in question [15] [16].”

After this sentence I would also introduce immersive Virtual Reality (iVR) which allows for more ecological valid studies to be conducted. Indeed, this added level of immersion adds to the realism of the environment, providing a more accurate representation of the space being tested.

Table 2

I would add “computer” in Concept 1 keyword as sometimes you can find (especially in older literature) something like “computer-based test”

Table 3

It seems search #2 is missing the keyword “preclinical”

Table 5

I would add an item which represent a category for the type of VR implied (desktop/semi/immersive)

I have doubts about the definition of item 5. If I understand correctly item 5 would be used to outline if the study has tested also other population beside the inclusion criteria (preclinical AD) such as people diagnosed with AD. However, given the focus on the preclinical population and its importance as an inclusion criteria, I think it would be more helpful to add a separate item specifically for the preclinical population. This item could then be used to categorize how "preclinical AD" was assessed, such as by genetic risk and type, family history, or any other scoring methods.

General comments

1. I like the wording of the types of VR implying the three types (desktop, semi, immersive). I think “semi” could be extended and by consequence an additional keyword in Table 3 1st row to AR (Augmented Reality). AR can be considered as specific class of “semi-immersive” and is gaining traction for being use as an assessment tool as the most ecological between all setups.

2. I don t think that there are many studies looking at preclinical AD so far, so I would be very cautions with keywords and exclusion criteria. For example in Table 2 Concept 3 I would add also “evaluating”, “testing”, “analysing”. Connected to this Table 4 exclusion criteria might be too strict. I would simplify by removing point 2 and 4 and I would make sure to include synonyms to the word “screening” for point 3. My concern is excluding a study that think of VR as a very secondary part of the study and thus mention the technique only in the methods.

7. PLOS authors have the option to publish the peer review history of their article (what does this mean?). If published, this will include your full peer review and any attached files.

Reviewer #1: No

Reviewer #2: **Yes: **

---

## [Author Response · Author response to Decision Letter 1]

10 Jan 2023

For your information, the response has been put down in the document "Response to Reviewers". Please kindly have a look at it. Thank you.

---

## [Decision Letter · Decision Letter 2]

15 Feb 2023

The use of virtual reality in screening for preclinical Alzheimer’s disease: a scoping review protocol

PONE-D-22-23396R2

Dear Dr. Tian,

We’re pleased to inform you that your manuscript has been judged scientifically suitable for publication and will be formally accepted for publication once it meets all outstanding technical requirements.

Kind regards,

Cosimo Ieracitano

Academic Editor

PLOS ONE

Additional Editor Comments (optional):

Reviewers' comments:

Reviewer's Responses to Questions

**Comments to the Author**

1. Does the manuscript provide a valid rationale for the proposed study, with clearly identified and justified research questions?

Reviewer #1: Yes

Reviewer #2: Yes

2. Is the protocol technically sound and planned in a manner that will lead to a meaningful outcome and allow testing the stated hypotheses?

Reviewer #1: Yes

Reviewer #2: Yes

3. Is the methodology feasible and described in sufficient detail to allow the work to be replicable?

Reviewer #1: Yes

Reviewer #2: Yes

4. Have the authors described where all data underlying the findings will be made available when the study is complete?

Reviewer #1: Yes

Reviewer #2: Yes

5. Is the manuscript presented in an intelligible fashion and written in standard English?

Reviewer #1: Yes

Reviewer #2: Yes

6. Review Comments to the Author

You may also provide optional suggestions and comments to authors that they might find helpful in planning their study.

Reviewer #1: Changes to satisfy the requested revisions. The clarification on what constitutes VR/preclinical AD is important. I would like to see a more rigorous definition of VR that doesn't perpetuate the confusion that VR includes desktop psychophysics but feel I might be a minority here.

Reviewer #2: The authors have replied to all of the comments mande and made adjustment accordingly when needed.

I have no more comments to add.

7. PLOS authors have the option to publish the peer review history of their article (what does this mean?). If published, this will include your full peer review and any attached files.

Reviewer #1: No

Reviewer #2: **Yes: **Andrea Castegnaro

---

## [Editor Report · Acceptance letter]

20 Feb 2023

PONE-D-22-23396R2 

The use of virtual reality in screening for preclinical Alzheimer’s disease: a scoping review protocol 

Dear Dr. Tian:

I'm pleased to inform you that your manuscript has been deemed suitable for publication in PLOS ONE. Congratulations! Your manuscript is now with our production department. 

Kind regards, 

on behalf of

Dr. Cosimo Ieracitano 

Academic Editor

PLOS ONE